# Effects of Floral Characters on the Pollination Biology and Breeding System of *Iris setosa* (Iridaceae): A Cold-Tolerant Ornamental Species from Jilin Province

**DOI:** 10.3390/biology14010002

**Published:** 2024-12-24

**Authors:** Xiyue Zhang, Ruoqi Liu, Lifei Chen, Tianhao Pei, Yu Gao, Xi Lu, Yunwei Zhou

**Affiliations:** 1College of Forestry and Grassland Science, Jilin Agricultural University, 2888 Xincheng Street, Changchun 130118, China; zxyiris123@163.com (X.Z.); 18843262894@163.com (R.L.); 2College of Horticulture, Jilin Agricultural University, 2888 Xincheng Street, Changchun 130118, China; zexichen@163.com; 3College of Plant Protection, Jilin Agricultural University, Changchun 130118, China; 20231486@mails.jlau.edu.cn (T.P.); gaothrips@jlau.edu.cn (Y.G.)

**Keywords:** flowering characteristics, breeding system, flower-visiting insects, pollination, flower development

## Abstract

*Iris setosa* is a cold-tolerant ornamental species, whose pollination system remains vague. Therefore, the authors aimed to determine the impact of flower characteristics on pollination biology as well as on the breeding system. In this paper, the authors conducted an in-depth study on the phenological period and flowering dynamics, floral organ characteristics, pollen viability and stigma receptivity, breeding habits, and flower-visiting insects of *I. setosa*. Furthermore, our study project shows a high adaptive evolutionary trend between floral morphological characters and pollination biology of *I. setosa* species.

## 1. Introduction

Plants produce flowers until visitation yields the amount of pollen necessary to produce the maximum number of seeds that resources can support [1]; thus, plants are simultaneously limited by resources and pollen. The frequency and potential consequences of pollen limitation for plant populations and communities have been intensely explored during the last few decades [2,3]. The pollinator limitation of female reproduction occurs when an inadequate supply of pollen limits the fruit set below the level possible given the plant’s available resources. A decrease in pollinator frequency (i.e., number of visits) likely decreases the quantity of pollen deposited onto the stigma [4], reducing the fruit and seed sets [5,6,7]. In addition, the plant’s ability to attract pollinators via flower morphological features can be crucial to fitness [8]. The limitation of fruit and seed production due to insufficient pollinator visitation is common and ubiquitous across plants [9].

China is one of the distribution centers of *Iris*. Despite being the largest genus in the family Iridaceae, with more than 250 species [10], there is little information on the role that pollinators play in the reproductive success of *Iris*. *Iris setosa* is a rhizomatous geophyte that belongs to the genus *Iris* and is widely distributed in northern temperate zones, such as North America and East Asia [11]. The inner perianth of *I. setosa* is small and almost invisible compared with other species of the family. The derived features of Iridaceae include two morphological characteristics: a unifacial isobilateral leaf and the presence of only three stamens [12]. Over time, the inner and outer whorls of the tepals have become progressively differentiated in *Iris*. Stripes, dots, or color gradients on the outer tepals produce nectar guides, a distinct feature intended to attract pollination insects to the flowers. In addition, *Iris* is a typical nectar plant. The nectaries are typically found between the base of the corolla tube and the base of the style, or they are spread across the inner wall of the corolla tube [13]. Most known *Iris* species have nectaries, such as *Iris sibirica*, which uses nectar guides to attract insects [14,15]. Many species are primarily attracted by floral resources and perianth pigments, with various flower fragrances serving as an attractant. For some pollinators, however, flower physical characteristics, particularly functional floral symmetry, can be just as significant.

The intricate flowers of the Iridaceae family are finely adapted for pollination by various animals, including hummingbirds, sunbirds, beetles, butterflies, moths, wasps, and bees. This intimate connection between flower form and pollination biology reveals how the wide range of flower colors, shapes, and scents are vital to the lives of the species [16]. Hymenoptera (mainly bees) pollinate the majority of Iridaceae plants. For example, bumblebees are one of the main pollinators of *Iris lutescens* [17,18]. Plants depend on one insect species or a small number of ecologically related species for pollination [19]. The diversity of floral characteristics of different *Iris* species leads to different attraction behaviors of pollinators. For example, the loose corolla of *Iris tectorum* is compatible with the pollination behavior of the large insects *Xylocopa appendiculata* and *X. tranquebarorum* [20]; *Gladiolus* (Iridaceae) are primarily or exclusively pollinated by flies with elongated mouthparts [21]; the corolla of salty *Iris halophila* is more tightly packed, and, therefore, the only effective pollinators are *Anthophora* sp. [22]; and *Iris pseudacorus* has narrower pollination pathways and attracts pollinators such as *Episyrphus balteatus*, which are flatter in size [23]. Pollinator limitation has a substantial impact on the reproductive success of cross-pollinated *Iris* species [24]. Cross-pollination of *I. sibirica* is carried out by insects, mainly bumblebees, honeybees, and male solitary bees, and, less often, by wasps and ants [25]. This ensures the success of reproduction. Some Iridaceae plants are known to be pollinated by bees and wasps, apparently seeking nectar rewards, while others are pollinated by beetles and flies, apparently seeking carrion for brood sites or food [26].

In this paper, we measured observations of local floral visitors to better understand *I. setosa*’s pollination techniques. The impacts of floral morphology and reproductive systems on pollination were studied to better understand how these elements influence reproductive success. This research aimed to (i) characterize floral resources foraged by visitors and pinpoint the most important group of pollinators based on their foraging behavior and frequency of visits; (ii) perform pollination experiments simulating the high frequency of visits of pollinators across flowers, assess the reproductive success between treatments by comparing fruit sets, and follow the plant population throughout the entire season to describe its phenology; and (iii) define the breeding system of *I. setosa*.

## 2. Materials and Methods

### 2.1. Study Sites and Species

*Iris setosa*, is in the Iridaceae. The experiment was conducted from April to July 2023–2024 in Jilin Agricultural University, China.

### 2.2. Observations on Floral Phenology and Morphology

We observed flowering phenology for individual flowers and entire plants in April–July 2023–2024. The flowering of 87 *I. setosa* populations was defined as three stages (10% of plants flowering at the same time was the first flowering period, 50% of plants flowering at the same time was the full flowering period, and less than 10% still flowering was the last flowering period). On the first day of flowering, three out of 10 randomly selected plants with similar growth were used to measure the length and width of 30 flowers in terms of flower diameter, perianth segments, filaments, anthers, pistils and ovaries of 30 flowers were measured with vernier calipers.

### 2.3. Determination of Pollen Viability and Stigma Receptivity

Selected unopened buds were marked, and anthers and stigmas from five flowers were taken at day before bloom, first day of bloom, second day of bloom, third day of bloom. TTC (2, 3, 5-Triphenyltetramlium chloride, Beijing, China) staining was used to test pollen viability on various flowering days, and pollen viability was computed using pollen coloration rate [27]. Stigma receptivity of *I. setosa* was detected using the benzidine-hydrogen peroxide (Beijing, China) method and examined under a microscope. The results showed that the stigma was strongly receptive if a significant number of air bubbles developed surrounding it [28].

### 2.4. Determination of Pollen Viability at Different Storage Temperatures

In mid to early June 2023, the soon-to-be-split *I. setosa* anthers were collected in four parts. We loaded them into 5 mL centrifuge tubes with caps and sealed with a small amount of silica gel desiccant. They were stored at room temperature 25 °C, 4 °C, −20 °C and −80 °C. After 1, 3, 5, 7, 9, 11, 13 and 365 days from the date of storage, five anthers were extracted from each of the four different temperature treatments. TTC staining was performed to determine pollen viability and calculate pollen storage life. In the laboratory, *I. setosa* anthers were placed on sulfate paper to disperse pollen naturally. We gently flicked the anthers with forceps to collect the scattered pollen and placed on a glass slide. We added 1–2 drops of TTC stain with a rubber-tipped dropper and mixed well with a dissecting needle. We covered the slide and stain for 20 min, protected from light. Then, we observed in the low magnification of a light microscope. Pollen grains with high viability were dark red, pollen grains with low viability were light red, and pollen without viability was colorless.

### 2.5. Estimation of Outcrossing Index (OCI) and the Pollen-Ovule Ratio

The Outcrossing index (OCI) was estimated using the criteria suggested by Dafni. A [29], as followers: (1) Flower diameters ≤ 1 mm were recorded as 0, 1–2 mm as 1, 2.1–6 mm as 2, and >6 mm as 3. (2) Simultaneous maturation of both pistil and stamen, or pistil first, was recorded as 0, and stamen first, as 1. (3) Pistil and stamen are recorded as 0 for the same height and 1 for herkogamy. When OCI is 0, the breeding habit is cleistogamy. When OCI is 1, the breeding habit is self-pollination. When OCI is 2, the breeding habit is facultative selfing. When OCI is 3, the breeding habit is self-compatibility and pollinators are sometimes required. When OCI is greater than or equal to 4, the breeding habit is mainly outcrossing, with some self-compatibility and need pollinators.

In *I. setosa* flower buds, randomly selected stamens were immersed in 1 mol·L^−1^ HCl (Guangzhou, China). The anther walls were removed by hydrolysis in a water bath at 60 °C for 1 h. Distilled water was added to the stamen suspension and the volume was fixed to 10 mL. 10 μL of the suspension was pipetted onto a slide, and the amount of pollen per stamen was observed and measured microscopically. The pollen count per stamen was 1000 by multiplying the mean value X. The ovaries of randomly selected *I. setosa* flower buds were dissected through the carpels with a dissecting knife under a stereomicroscope (Leica DM2500, Wetzlar, Germany) 20 times, the number of ovules was counted and recorded, and the average value was computed. The type of *I. setosa* breeding system was evaluated using Cruden RW’s criterion [30].

### 2.6. Sampling and Calculation of Nectar Characteristics

Flowers that opened on the same day were chosen to assess the nectar content and nectar sugar concentration in their natural state Wang [31]. Ten plants were chosen, and two neighboring flowers were chosen at random from each plant, yielding a total of 20 blooms for measurement. Flowers were chosen based on the following criteria: they had to be freshly opened and show no evident signs of insect feeding. Measurement of 24 h storage capacity: the bud-bearing meristems of *I. setosa* were bagged with different sizes of mesh bags, and nectar traits were measured on the first day after the flowers opened, and the nectar volume and sugar content of entire flowers were measured and estimated.

Nectar volume was measured by aspirating all nectar secreted by the nectar glands using a disposable capillary tube (5 μL/32 mm, ringcaps^®^, Hirschmann, Germany)and measuring the length of the column with a vernier caliper.

Sugar concentration was measured by aspirating nectar using a 5 μL capillary tube and measuring it with a hand-held glucometer (0–32% Brix, LICHEN, Shanghai, China). The temperature was 23.9–28.5 °C. The nectar volume is calculated by converting the length of the measured nectar column (L) to the nectar volume (μL, the length of a 5 μL capillary tube is 32 mm) using the following formula:Nectar volume = *L* × 5/32(1)
Y = 0.00226 + 0.00937X + 0.0000585X^2^
(2)

The sugar content of nectar (Z, mg) was calculated from the measured values of nectar volume, where X is the nectar sugar concentration (%) and Y is the sugar content of nectar per microliter (mg/μL), using the following equation [32]:Z = *V* × Y (V is the nectar volume, μL) (3)

### 2.7. Sampling and Preservation of Pollen Tubes

We used fluorescence monitoring to observe pollen tube germination and growth in natural conditions, as well as artificial self-pollination and artificial cross-pollination of *I. setosa*. *Iris setosa* styles and ovaries were collected at 1, 2, 4, 6, and 8 h after pollination. Four samples were taken at each time point and kept in FAA (Fuzhou, China) fixative at 4 °C.

The fixative was poured out and rinsed with distilled water 2–3 times before observation. 1 mol·L^−^^1^ NaOH (Tianjin, China) was added and placed in a 60 °C water bath to soften for about 1 h. 1% Aniline Blue (Water Soluble) was used for staining, and after staining for 24 h at room temperature under dark conditions, the samples were slowly placed on slides and the coverslips were pressed. The samples were observed under an Olympus microscope (Leica DM2500, Wetzlar, Germany) and photographed.

### 2.8. Observations on Flower Visiting Insect Species and Behavior

In order to understand the types of *I. setosa* flower visitors and their collection behaviors of pollen and nectar, the collection behaviors of different flower visitors were observed on sunny days in June 2023–2024 at 08:00–11:00 a.m. and 12:00–17:00 p.m. Two plots (1 × 1 m) were randomly established with approximately 30 flowers per plot. These plots were observed for 60 min per day for a total of 28 observations. A camera (Canon EOS 70D, Tokyo, Japan) was used to record the foraging behavior of several floral visitors. Furthermore, the number of flower visits and the number of flowers visited by different flower visitors per 30 min were recorded, and each flower visitor’s visit rate (the number of times a flower was visited every 30 min) was calculated by dividing the total number of flowers observed by the number of flowers visited in each 30-min period. The primary pollinators were identified based on the frequency of flower visits and foraging behavior. Bees bearing pollen masses were caught and identified as prospective pollinators based on whether the insect touched the stigma’s pollinable surface during flower visits, resulting in cross-pollination.

### 2.9. Artificial Pollination Test

Pollinator limitation studies were conducted on *I. setosa* populations in 2023 and 2024 by comparing hand-pollinated and naturally pollinated flowers. Potential animal pollinators or seed dispersers were identified through in-person observation and video camera use. The treatment is as follows:

T1 is control (CK) natural pollination. T2 treatment is direct bagging of flower buds to detect the need for pollinators in the natural state. T3 treatment is removal of males with forceps before flowering and bagging to detect the presence of apomixis. T4 treatment is removal of males with forceps before flowering without bagging to detect the extent to which pollinators contribute to seed set in the natural state. T5 treatment was to remove the male with forceps before flowering, set a bag, and artificial self-pollination on the next day to detect the presence of self-compatibility. T6 treatment was to remove the male with forceps before flowering, set a bag, and artificial geitonogamy on the next day. T7 treatment was to remove the male with forceps before flowering, set a bag, and artificial xenogamy on the next day. Each treatment group needed 20 flowers, bagged immediately after treatment, and the paper bag could be removed after 15–20 d. Fruit set rate, seed number and seed set rate of each group were counted.

### 2.10. Statistical Analysis

Data from this experiment were statistically analyzed using SPSS 26.0 (IBM Corp., Chicago, IL, USA) statistical analysis software. One-way ANOVA was used to analyze whether there were significant differences in the number of seeds and seed set between the different treatments of the pollination experiment.

## 3. Results

### 3.1. Flowering Characteristics and Flowering Period Characteristics

As shown in Figure 1, the average diameter of *Iris setosa* flowers was approximately 67.9 mm (Table 1). The outer tepals were three, approximately 45.1 mm long and 23.1 mm wide, with a pattern of blue-purple pinstripes at their bases as nectar guides (Figure 1A). The inner tepals were three, approximately 10.1 mm long and 4.5 mm wide (Figure 1B). The stamens were three, approximately 20.3 mm high, growing close to the stylar branches. The anthers were purple, outwardly dehiscent, and situated on the outer sides of the styles, just underneath the stigmas. The filaments were as long as the anthers and white with a purple halo. The stylar branches were flattened and arching with a pronounced bend (Figure 2). The style color was lighter than the tepals, forming a semi-enclosed passage with an outer perianth for pollinating insects to enter. The height was ca. 29.9 mm, the apical lobes were nearly square, and the margins were sparsely dentate. Three stigmas were located at the base of the apical lobes of the style, with a height of ca. 2.2 mm. The stigmas folded upward at maturity (Figure 2). The nectaries were located in the region between the base of the corolla tube and that of the style. The ovary was inferior, ca. 11.7 mm high, long-cylindrical, yellowish-green, and three-loculed, with median placentation and numerous anatropous ovules (Figure 1C).

*Iris setosa* began to sprout into the nutritive growth stage in late April in 2023–2024 (Figure 3A,B), entered the stem extraction stage in mid-May (Figure 3C), and began to show buds in late May (Figure 3D). In 2024, the initial flowering period was from 28 May to 1 June (Figure 3E), the blooming period was from 2 June to 13 June (Figure 3F), and the end flowering period was from 14 June to 23 June (Figure 3G). In 2023, the initial flowering period was from 25 May to 30 May, the blooming period was from 31 May to 11 June, and the end flowering period was from 12 June to 20 June (Figure 3G). Since the average temperature in May–June in 2023 was 3–4 °C higher than that in 2024, the overall flowering period was 2–3 days earlier than that in 2024. During single-flower opening, the buds gradually expanded (Figure 4A,B), and the outer perianth dispersed one by one, with the apical part of the perianth drooping until all three outer perianths drooped (Figure 4C–E). After 2–4 d of flowering, the whole perianth curled up, the stamens dried up (Figure 4) and gradually withered (Figure 4G–J), and the ovary of the pollinated flower began to expand (Figure 4K,L). The flowering periods of single flowers and single plants were 2–4 d and 10–14 d, respectively.

### 3.2. Pollen Viability and Stigma Receptivity

Pollen viability is an important parameter for evaluating pollen quality. As shown in Table 2, TTC (2, 3, 5-triphenyltetramlium chloride) staining experiments were conducted on flowers collected on the day before bloom, the first day of bloom, the second day of bloom, and the third day of bloom. The results show that *I. setosa* pollen was already highly viable on the day before flowering (Figure 5). The first day of flowering had the highest viability of (96 ± 2)%, which was the best time for pollination. Overall, the viability showed an increasing and then decreasing trend (Table 2).

*Iris setosa* stigmas were not receptive on the day before flowering (Figure 6 and Table 2). The stigmas were most receptive on the first day of flowering. Thereafter, stigmatic receptivity declined over time and could last until the third day of flowering, suggesting that *I. setosa* exhibits dichogamy.

### 3.3. Effect of Different Storage Temperatures on the Determination of Pollen Viability

As shown in Figure 7, the pollen viability rate of *I. setosa* pollen was 62% on the first day at room temperature (25 °C), and pollen viability was lost after 24 h. The lifespan of *I. setosa* pollen could be prolonged under the three preservation conditions of low-temperature storage (4 °C), low-temperature freezing storage (−20 °C), and ultra-low-temperature freezing storage (−80 °C), which could last approximately 24 h, at most, under room-temperature storage conditions. The figure shows that the pollen viability rate of *I. setosa* under cryopreservation, cryogenic freezing, and ultra-low-temperature freezing did not significantly differ at the start of the preservation period but steadily decreased thereafter. The viability rate of *I. setosa* pollen did not significantly differ from that of cryopreserved and ultra-low-temperature-frozen pollen at the beginning of the storage period. The viability rate of cryopreserved and cryofrozen pollen gradually decreased with the increase in the storage time. Moreover, the viability rate of the pollen in cryopreserved and cryofrozen conditions decreased more slowly than that under other storage conditions. The longest storage time was 365 days, and the average pollen viability rate reached 74% in 365 d.

### 3.4. Outcrossing Index and Pollen-Ovule Ratio

The pendant petals of the corolla of *I. setosa* are the most enlarged part of the flower and can provide a basis for measuring the size of the floral organ. The length and width of the pendant petals were about 45 mm and 23 mm, respectively (Table 1), and were larger than 6 mm. Additionally, the outcrossing index (OCI) was recorded as three, and the stamen-first was recorded as one. The anthers were spatially separated from the stigma, and the OCI was recorded as one. In summary, the OCI of *I. setosa* was five, indicating that the sexual reproduction system of *I. setosa* was mainly outcrossing, with some self-compatibility and need for pollinators. The average pollen number of single flowers of *I. setosa* was 34,200, the average ovule number was 88, and the pollen-ovule ratio (P-O) was 388.6, indicating that the sexual breeding system of *I. setosa* is facultative selfing or facultative xenogamy.

### 3.5. Nectar Characteristics

The calculation of the nectar volume and sugar content of *I. setosa* on the day of flowering in the natural state showed that the nectar volume was 19.6 ± 4 μL. The sugar concentration and content of the nectar were 17% and 3.5 ± 0.7 mg, respectively.

### 3.6. Fluorescence Microscopy of Pollen Tube Growth

The fluorescence observation of pollen tube growth was carried out on *I. setosa* after natural flowering and *I. setosa* after artificial self-crossing and cross-pollination, respectively. The results are shown in Figure 8, Figure 9 and Figure 10.

The growth of pollen tubes after the natural flowering of *I. setosa* is shown in Figure 8. When *I. setosa* naturally flowered for 0 d in the morning, pollen was observed on the stigma of the pistil. At this time, the anthers dispersed the pollen completely, probably due to the insects feeding on the nectar or pollen, transferring the pollen they carried to the stigma (Figure 8A). By the afternoon, we observed pollen sprouting out of the pollen tubes and growing in bundles (Figure 8B). Based on the previous section, *I. setosa* pollen was most active on the day of flowering, prompting pollen tubes that continued to extend downward. By 1 d of natural flowering in the morning and 1 d in the afternoon, the pollen tubes reached the vicinity of the ovule. Subsequently, the pollen tubes entered the embryo sac to complete fertilization (Figure 8C,D).

The results of the fluorescence observation of pollen tube growth after self-pollination demonstrate the following: After self-pollination for 1 h, *I. setosa* pollen sprouted pollen tubes on the stigma (Figure 9B). At 2 h of self-pollination, the pollen tubes grew in bundles (Figure 9C), and the pollen tube growth trajectory was irregular. A small number of pollen tubes were observed sprouting and entangled with each other in clusters after 4 h of pollination (Figure 9D), and pollen tube anomalies appeared. Pollen tubes continued to grow downward in the style (Figure 9E,F). A small number of pollen tubes appeared to be bent and broken within the channel (Figure 9G,H). After 6 h of pollination, the pollen tubes reached the ovule (Figure 9I). Hence, the pollen tubes took 4–6 h to cross the style, and no substances that inhibit pollen tube growth were present in the style.

The results of the fluorescence observation of pollen tube growth after cross-pollination demonstrate the following: After cross-pollination for 1 h, *I. setosa* pollen sprouted pollen tubes on the stigma (Figure 10B). At 2 h of cross-pollination, the pollen tubes grew in bundles (Figure 10C). Numerous pollen tubes were observed to sprout and twist back and forth with each other 4 h after pollination (Figure 10D), and the pollen tubes continued to grow downward and appeared to bend (Figure 10E,F). The pollen tubes entered the bottom of the style and reached the ovule 6 h after pollination (Figure 10G,H). After 8 h of pollination, many pollen tubes wrapped around the ovule (Figure 10I).

In summary, the insect vector can pass *I. setosa* pollen to the pistil stigma under natural conditions. However, pollen viability is an important prerequisite for pollen tube growth. The pollen sprouts pollen tubes only when viable. These pollen tubes gradually extend downward to the ovary ovule to complete fertilization. Then, the *I. setosa* plant can bear fruits after natural flowering. In the fluorescence observation of *I. setosa* after self-pollination, abnormal phenomena, such as pollen tube folding and twisting, breaking and bending, and dried pollen grains not sprouting, were found, which made it difficult to complete the fertilization of the pollen tubes and reduced the fruiting rate of *I. setosa* after self-pollination.

### 3.7. Observations of Flower-Visiting Insect Species and Behavior

Photographs of flower-visiting insects were taken during the experimental observations. As shown in Figure 11, the species of flower-visiting insects were mainly *Apis mellifera* (Figure 11A), *Megachile* sp. (Figure 11B), *Syrphus corollae* (Figure 11C), *Episyrphus balteatus* (Figure 11D), *Lasioglossum* sp. (Figure 11E), and others. Small numbers of flower-visiting insects such as Mordellidae (Figure 11F), ants, and beetles were also observed, alongside a small number of spiders, which may be predators of flower-visiting insects.

Insect flower-visiting behavior is related to the characteristics of the flower itself. *Iris* is a typical insect-pollinated flower. The species, number, and frequency of insect visits are important factors affecting the fruiting rate. Apoidea and Syrphidae were the main pollinators of *Iris*. The peak time of flower visits was from 10:00 to 14:00 every day. We did not observe any insects within flowers during morning observations on rainy or windy days. The time taken by *Apis mellifera* to visit a flower in sunny weather ranged from 3 to 17 s. The number of visits to flowers in 30 min was 112 (Table 3). When the insects entered the pollination channel between the stigma and the outer perianth to feed on pollen or to suck up nectar in the corolla tube of the upper ovary, they carried a large amount of pollen via the hairs on their thoraxes and abdomens, and a large amount of pollen mass adhered to their legs (Figure 12A–C). The pollination was accomplished by touching the stigma of another flower during the next pollen or nectar collection.

*Lasioglossum* sp. visited a flower for 8–28 s under clear weather conditions and visited 32 times in 30 min. They entered the pollination channel upside down when collecting pollen, and the pollen grains adhered to their head and legs could touch the stigma for pollination (Figure 12D).

*Syrphus corollae* visited a single flower for 45 s^−1^ min 7 s on a sunny day, and the number of flower visits was 10 in 30 min. Their heads and antennae contacted the stigma first when feeding on pollen. However, their legs did not adhere to the excess pollen mass during the visits due to having fewer downy hairs on their bodies compared with bees (Figure 12E,F).

*Episyrphus balteatus* visited a single flower for 24–32 s on a sunny day and six times in 30 min. They were small in size, and their heads touched the stigma first when burrowing into the pollination channel to enter the flower to feed on pollen (Figure 12G).

*Megachile* sp. visited a single flower for 4–16 s on a sunny day, and the number of flower visits was 106 in 30 min. Its mouthparts were longer than those of other honeybees to suck nectar directly. The leaf-cutting bees entered the pollination channel directly from the front side or by holding the styles on the reverse side during the flower visit. The tomentum on their abdomens adhered to the pollen dispersed on the outer perianth during frontal visits (Figure 12H,I). On the next visit to the flower, they entered the channel by holding the style on the opposite side, and the pollen adhering to their abdomens contacted the stigma for pollination. A few flower-visiting insects, such as flower fleas, ants, and spiders, bit the corolla tube and sucked the nectar, but no pollen was collected.

### 3.8. Artificial Pollination Test Results

As shown in Table 4, seven different pollination treatments had significant effects on the fruit set, seed set, and seed number of *I. setosa*. The fruit set, seed set, and seed number of T2 and T3 were all zero, indicating that *I. setosa* could not self-pollinate nor be subjected to apomixis. The seed set in the T4 group was significantly different from that in T1. The fruit set in T6 and T7 was in the range of 95% and 100%, respectively, and the number of seeds and seed set were not significantly different from those in T1. Meanwhile, the number of seeds and seed set were significantly lower in T5 than in T1, T6, and T7. The results of T6 and T7 were similar, which indicated that the breeding system of *I. setosa* is based on outcrossing with a certain degree of self-compatibility.

## 4. Discussion

Different species of *Iris* attract different pollinators because of their varying floral characteristics. For example, *Lapeirousia anceps* (Iridaceae), which has a bimodal distribution of floral tube lengths, and the long-proboscid fly *Moegistorhynchus longirostris* (Nemestrinidae), its main pollinating insect, have a unimodal distribution of proboscis lengths and show a preference for the long-tube phenotype [33]. The pollinators of *Iris pumila* showed a higher affinity for flowers with taller flower stems and greater brightness and size of floral organs [34]. In this study, Hymenoptera insects belonging to *Apis mellifera*, *Megachile* sp., and *Lasioglossum* sp. were common pollinators of *Iris setosa*. We did not observe any insects within flowers during morning observations on rainy or windy days.

The base of the outer perianth often has appendages called barbs or nectar guides with spots and veins, which can be used as visual signals for insects to land and enter the pollination channel [35]. The thin bluish-purple stripes that extend outward on the outer perianth of *I. setosa* are called nectar guides (Figure 1A). Most of the pollen is also dispersed on the outer perianth after the anther of *I. setosa* opens to attract insects. The structure and function of the outer perianth of *I. setosa* have been completely specialized during its long-term evolution process, which may be an important means of attracting a wide range of insect pollinators. It is hypothesized that there has been an adaptive evolutionary process between the floral features of *I. setosa* and the corresponding flower-visiting insects, which has led to a synergistic evolution.

The scent of the flower and the pollination rewards (pollen and nectar) attract different pollinating insects, which promotes pollination and cross-breeding and enhances the fertility and resilience of the offspring so that they can reproduce successfully and evolve over time [36]. The flower size and nectar guide could act as visual signals, where large flowers/patches indicate larger tunnels (where pollinators shelter), increasing the probability of fruits and seeds [37]. It was found that the nectaries of *I. setosa* were located in the area between the base of the style and the inner part of the corolla tube. Insects were most active on the day of flower opening, and the volume of nectar from a single flower was 19.6 ± 4 μL. It was hypothesized that the nectaries of *I. setosa* secrete a large amount of nectar on the day of flower opening, which attracts pollinators to visit the flowers. The frequency of flower visits and the amount of pollination reward were the main factors affecting the pollination efficiency [38]. *Apis mellifera* and *Megachile* sp. had the highest number of visits within 30 min, which were significantly higher than the other visiting insects, followed by *Lasioglossum* sp. During the observation, it was found that the bees had a large amount of pollen mass adhered to their legs, and the downy hairs on their bodies carried pollen. The amount of pollen adhered to the contact surface of the stigma was larger during the next visit to the flower, thus increasing the success rate of pollination. Furthermore, the seed-setting rate of *I. setosa* was higher under natural pollination conditions.

Spatial (herkogamy) or temporal (dichogamy) separation of sex organs is a mechanism thought to limit self-pollination and promote cross-pollination [39]. Artificial pollination and field observations have shown that only natural pollination or artificially pollinated flowers produce capsules, whereas self-pollination after bagging does not produce fruit. Flower reward (nectar and pollen) is the main means of attracting insects to plants [40]. The seed set rate obtained from group T4 (not bagged and emasculated) was significantly lower than that of group T1 (natural pollination). The main behavior of flower-visiting insects of *I. setosa* was to feed on pollen and nectar during the period of flower visitation. When pollen was not available as a pollinator’s reward, only nectar could be provided, and the number of flower visits by *Syrphus corollae* and *Episyrphus balteatus* was almost zero. The higher number of flower visits by Apoidea, and the presumed pollination reward of Syrphidae, which was mainly pollen (Table 3), reduced the attraction of flower-visiting insects, thus affecting the seed set. The number of seeds and the seed set did not significantly differ between T1 (natural pollination), T6 (artificial geitonogamy), and T7 (artificial xenogamy), which is presumed to be due to the high number of species and number of pollinators; therefore, there is no pollinator restriction. According to the ratio of pollen ovules of *I. setosa* and the OCI, the breeding systems were judged to be mainly outcrossing, with some self-compatibility and need for pollinators, whereas other Mediterranean *Iris* species are largely self-incompatible [41,42]. Therefore, domesticated *I. setosa* is suitable for planting in natural habitats where insects are more abundant and widespread, easier for natural sexual reproduction, avoids inbreeding decline, and is important for maintaining species diversity.

## 5. Conclusions

*Iris setosa* presents pollen and nectar as a reward to floral visitors. Successful sexual reproduction indicates that *Apis mellifera*, *Megachile* sp., and *Lasioglossum* sp. are its major effective pollinators due to the frequency of flower visits and the number of visits. The unique flower morphology of *I. setosa* facilitates cross-pollination, which attracts a variety of flower-visiting insects while also allowing it to adapt to these insects’ characteristics. In the process of flower visits, these insects complete pollen transfer and promote the effective reproduction of *I. setosa*. Moreover, the different pollination efficiencies and methods of the different insects provide a genetic diversity of *I. setosa* populations, which enables these populations to have stronger adaptive ability and survivability in the face of environmental changes.

## Figures and Tables

**Figure 1 biology-14-00002-f001:**
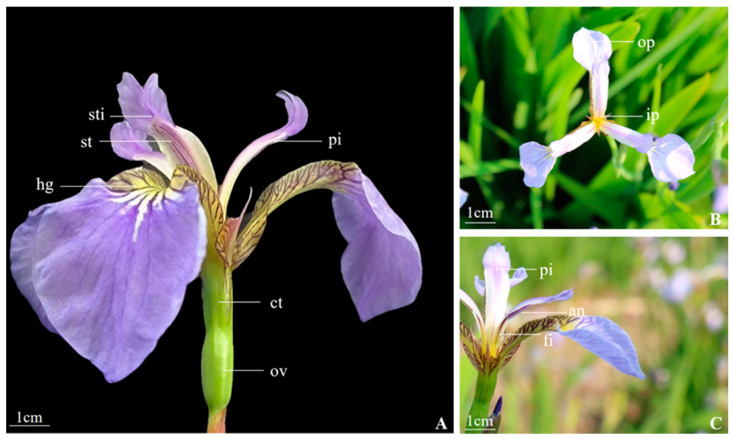
The characteristics of the flower of *I. setosa*. (**A**) The overall structure of the corolla of *I. setosa*, (**B**) top view of corolla, (**C**) floral unit. sti. Stigma, st. Stamen, pi. Pistil, hg. nectar guide, ct. Corolla tube, ov. Ovary, op. Outer perianth, ip. Inner perianth, an. Anther, fi. Filament.

**Figure 2 biology-14-00002-f002:**
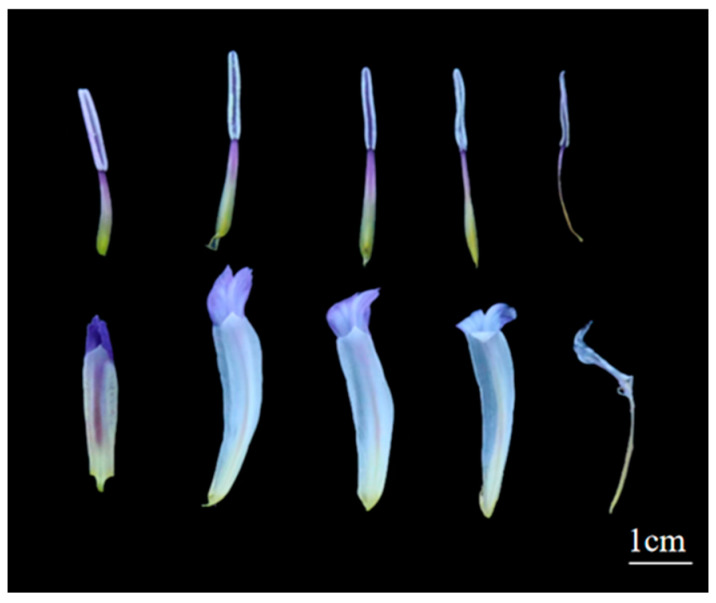
Changes of stamens and pistils in *I. setosa* during flowering. From left to right, 3 p.m. the day before bloom, 9 a.m. on the day of bloom, 3 p.m. on the day of bloom, 9 a.m. on the first day of bloom, and 9 a.m. on the second day of bloom.

**Figure 3 biology-14-00002-f003:**
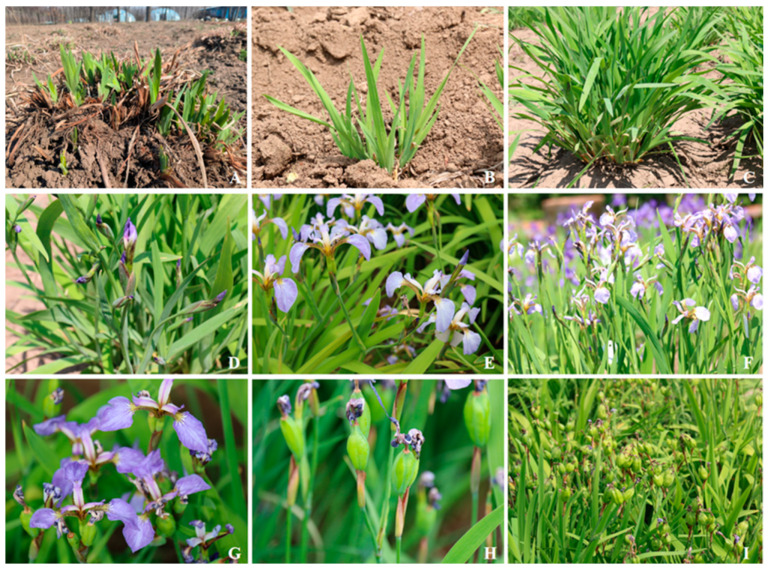
Annual phenology of *I. setosa* group. (**A**,**B**) nutritive growth stage, (**C**) stem extraction stage, (**D**) squaring period, (**E**) initial flowering period, (**F**) blooming period, (**G**) end flowering period, (**H**,**I**) fruit period.

**Figure 4 biology-14-00002-f004:**
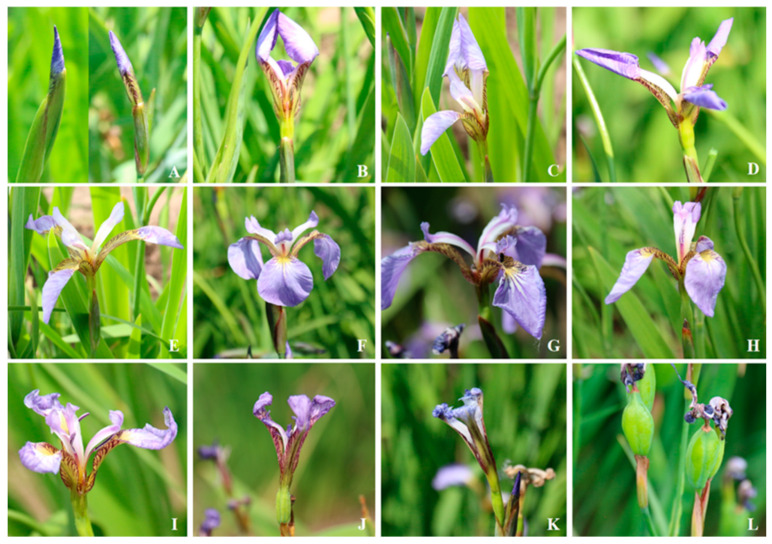
The flowering process of *I. setosa*. (**A**) bracts, buds, (**B**) buds about to open, (**C**–**E**) perianths are scattered one by one, (**F**) blooming flowers, (**G**–**J**) flowers gradually withered, (**K**,**L**) ovary enlarged.

**Figure 5 biology-14-00002-f005:**
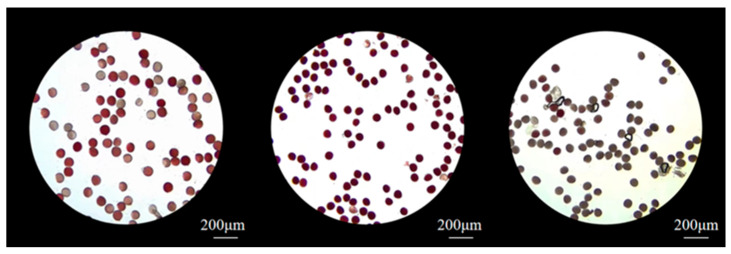
Pollen vitality of *I. setosa*.

**Figure 6 biology-14-00002-f006:**
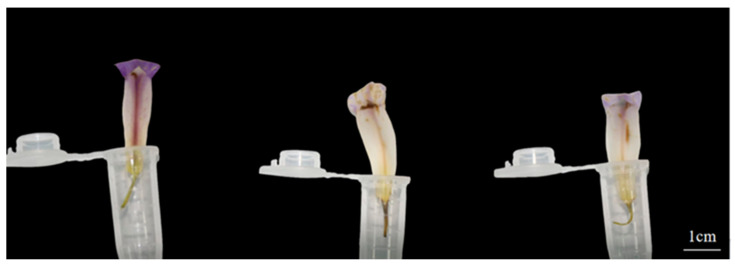
Stigma receptivity of *I. setosa*.

**Figure 7 biology-14-00002-f007:**
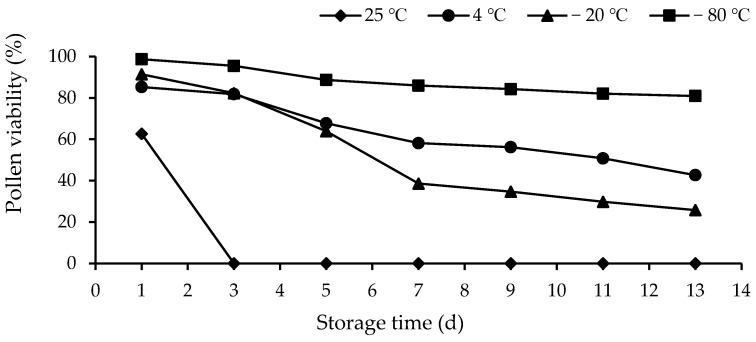
Changes in pollen viability of *I. setosa* under different storage temperatures.

**Figure 8 biology-14-00002-f008:**
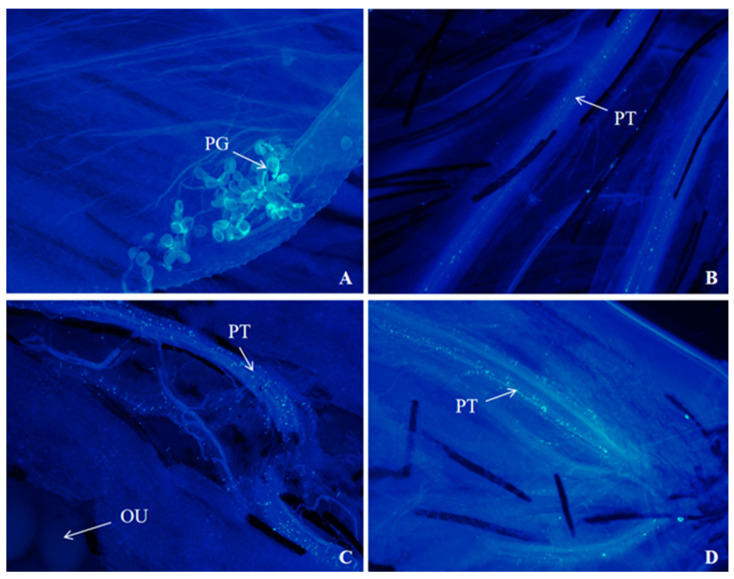
Fluorescence observation of pollen tube growth after natural flowering of *I. setosa*. (**A**) Natural flowering 0 d in the morning, (**B**) Natural flowering 0 d in the afternoon, (**C**) Natural flowering 1 d in the morning, (**D**) Natural flowering 1 d in the afternoon. PG: Pollen grain, PT: Pollen tube, OU: Ovule.

**Figure 9 biology-14-00002-f009:**
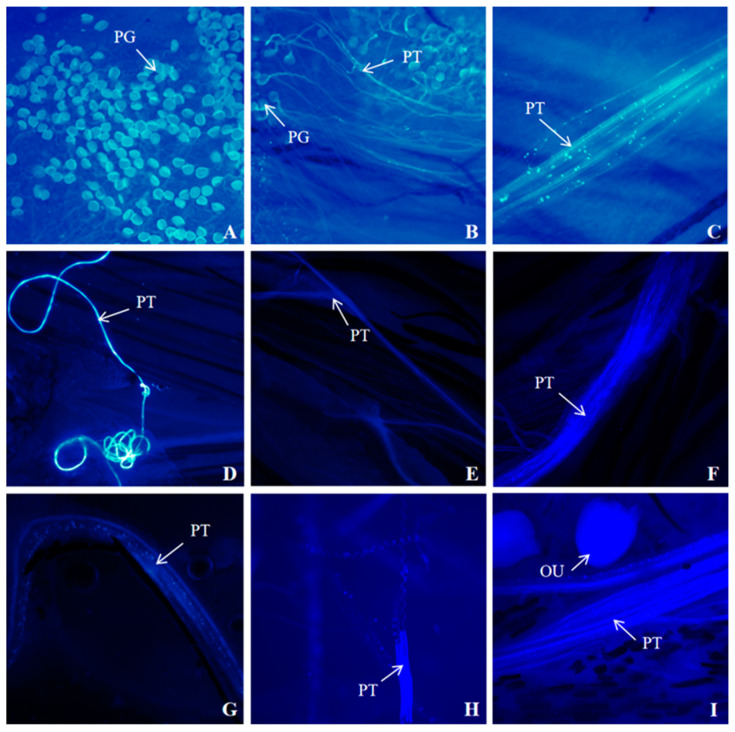
Fluorescence observation of pollen tube growth of *I. setosa* after artificial self-pollination. (**A**) Pollen morphology under fluorescence, (**B**) Pollen germinated on the stigma at 1 h after pollination, (**C**) Pollen tube bundle growth at 2 h after pollination, (**D**) Pollen tube folds intertwined at 4 h after pollination, (**E**) Pollen tubes entered the style, (**F**) Pollen tube continues to grow downward, (**G**) Pollen tube bending, (**H**) Pollen tube fracture, (**I**) Pollen tubes reaches the ovule at 6 h after pollination. PG: Pollen grain, PT: Pollen tube, OU: Ovule.

**Figure 10 biology-14-00002-f010:**
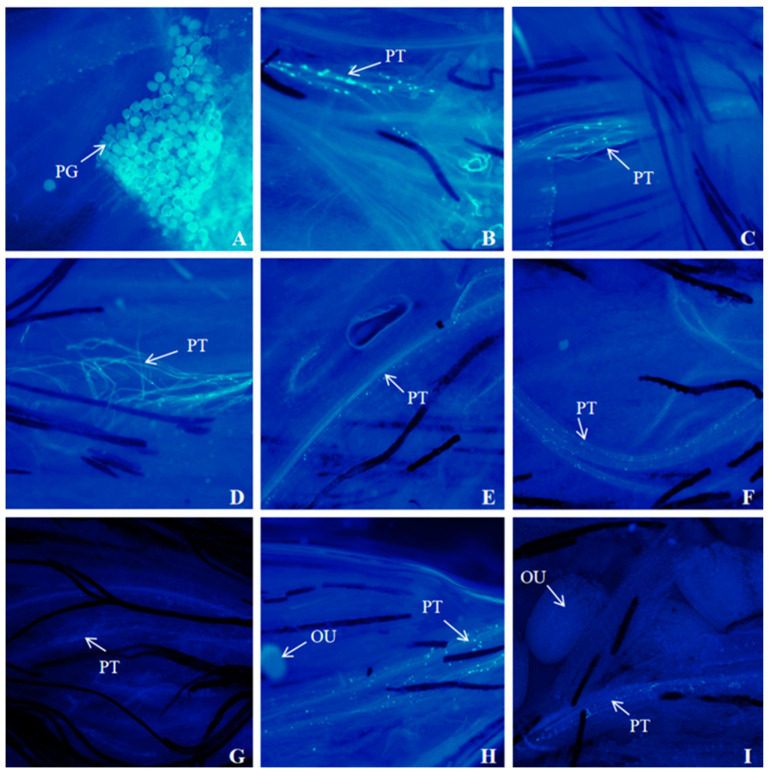
Fluorescence observation of pollen tube growth of *I. setosa* after artificial cross pollination. (**A**) Pollen morphology under fluorescence, (**B**) Pollen germinated on the stigma at 1 h after pollination, (**C**) Pollen tube bundle growth at 2 h after pollination, (**D**) Pollen tube folds intertwined at 4 h after pollination, (**E**) Pollen tube continues to grow downward, (**F**) Pollen tube bending, (**G**) Pollen tube reaches the bottom of the style, (**H**) Pollen tubes reaches the ovule at 6 h after pollination, (**I**) Pollen tube wrapped around the ovule at 8 h after pollination. PG: Pollen grain, PT: Pollen tube, OU: Ovule.

**Figure 11 biology-14-00002-f011:**
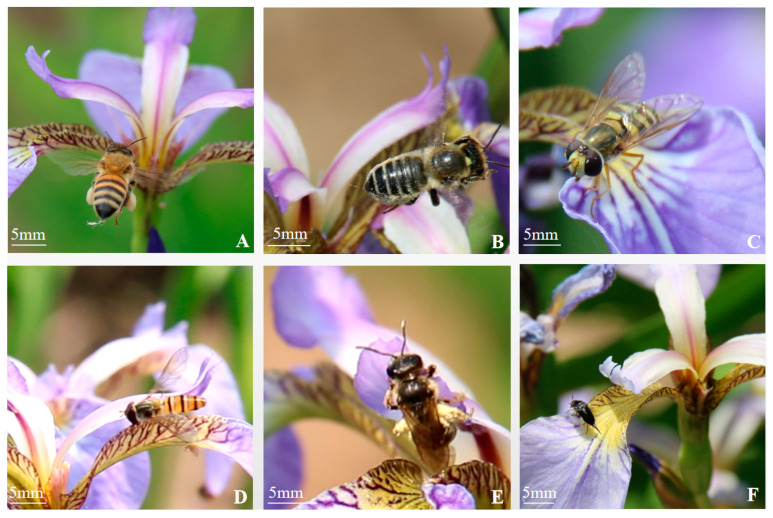
Flower visiting insect species. (**A**) *Apis mellifera*, (**B**) *Megachile* sp., (**C**) *Syrphus corollae*, (**D**) *Episyrphus balteatus*, (**E**) *Lasioglossum* sp., (**F**) Mordellidae.

**Figure 12 biology-14-00002-f012:**
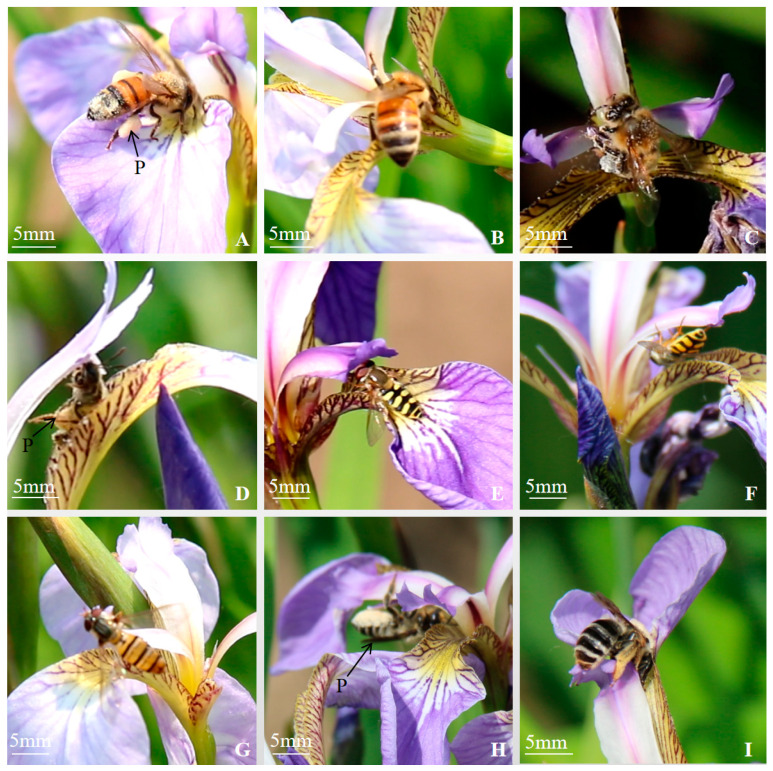
Behaviour of flower-visiting insects. (**A**–**C**) The pollination process of *Apis mellifera* (p. pollen ball), (**D**) Flower Visiting Behaviour of *Lasioglossum* sp. (p. pollen ball), (**E**,**F**) The pollination process of *Syrphus corollae*, (**G**) Flower Visiting Behaviour of *Episyrphus balteatus*, (**H**,**I**) The pollination process of *Megachile* sp. (p. pollen ball).

**Table 1 biology-14-00002-t001:** Floral parameters of *I. setosa*.

Floral Organ	mm Size
Outer perianth length	45.1 ± 3.8
Outer perianth breadth	23.1 ± 1.9
Inner perianth length	10.1 ± 0.6
Inner perianth breadth	4.5 ± 0.6
Ovary height	11.7 ± 1.8
Style height	29.9 ± 2.2
Stigma height	2.2 ± 0.4
Stamen height	20.3 ± 1.7
Flower diameter	67.9 ± 9.1

Data are mean ± SD.

**Table 2 biology-14-00002-t002:** Pollen viability and stigma receptive for *I. setosa*.

Time/d	Pollen Viability/%	Stigma Receptive
−1	91 ± 3	−
1	96 ± 2	+++
2	66 ± 9	++
3	29 ± 7	+

“−” indicates that the stigma is not receptive, “+” indicates that the stigma is receptive, “++” indicates that the stigma is highly receptive, “+++” indicates that the stigma is extremely receptive.

**Table 3 biology-14-00002-t003:** Visitors and their behavior on flowers of *I. setosa*.

Visitor	Visitation Rates	Number (%) of Visits	Visitor Type	Reward
*Apis mellifera*	0.438 ± 0.080 ^a^	112 (42)	Pollinator	Nectar, pollen
*Megachile* sp.	0.406 ± 0.087 ^a^	106 (40)	Pollinator	Nectar, pollen
*Lasioglossum* sp.	0.154 ± 0.039 ^b^	32 (12)	Pollinator	Nectar, pollen
*Syrphus corollae*	0.080 ± 0.058 ^c^	10 (4)	Pollen thief	pollen
*Episyrphus balteatus*	0.066 ± 0.052 ^c^	6 (2)	Pollen thief	pollen

Visits per flower per 30 min (mean ± SE). Different letters show significant differences at *p* < 0.05 (GLMs). Visits of insect species percentage of the total visits is indicated in bracket.

**Table 4 biology-14-00002-t004:** Results of Artificial Pollination Trial of *I. setosa*.

Pollination Test	Number of Flowers	Fruit Setting Rate/%	Seed Number/%	Seed Setting Rate/%
T1	20	90.00	85 ± 10 ^a^	63 ± 11 ^a^
T2	20	0.00	0 ± 0 ^d^	0 ± 0 ^d^
T3	20	0.00	0 ± 0 ^d^	0 ± 0 ^d^
T4	20	55.00	37 ± 23 ^c^	29 ± 14 ^c^
T5	20	85.00	59 ± 9 ^b^	43 ± 10 ^b^
T6	20	95.00	88 ± 5 ^a^	63 ± 9 ^a^
T7	20	100.00	86 ± 7 ^a^	65 ± 9 ^a^

T1 group natural pollination (CK), T2 group was directly bagged, T3 group emasculation and bagged, The T4 group not bagged and emasculation, The T5 group of artificial self-inbreeding, The T6 group of artificial geitonogamy, The T7 group of artificial xenogamy, Different lowercases in the same column indicate the significant (*p* < 0.05) difference.

## Data Availability

Data are contained within the article.

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
