# Peer review of "Effects of Floral Characters on the Pollination Biology and Breeding System of Iris setosa (Iridaceae): A Cold-Tolerant Ornamental Species from Jilin Province"

_biology, 2024, doi:10.3390/biology14010002_

Round 1

Reviewer 1 Report

Comments and Suggestions for Authors

The article is written on an urgent topic concerning the peculiarities of pollination of a rare species of iris. The study is full-fledged, the authors have applied a sufficient number of methods for conducting observations and analyzing the results obtained. the conducted studies correspond to the set goal, and the conclusions correspond to the results obtained. However, there are significant issues that require clarification and additions. keywords: it is necessary to remove the Latin name (it is already in the title of the article) and possibly replace it with a local or folk name, as well as add the words "pollination", "flower development". 

Section 1. Introduction - not enough information. It is necessary to add information about insect pollinators, as well as to compare the species of iris by type of pollination and the peculiarities of flower development.
In the Methodology section: 2.1. The authors obtained data for one year of research. one year is not enough to study the timing and duration of flowering, since these parameters depend on weather conditions, which in different years can significantly. For such studies, it is necessary to provide data for several years in comparison with the climatic conditions in the years of research. 2.2. The total number of plants is not specified, the sample of 3 plants is very small to assess quantitative characteristics.

Section 3 Results - in the presentation of quantitative indicators, there is no need for the accuracy of rounding to hundredths with initial measurements in millimeters. Also, for data presented as a percentage, it is enough to round up to integer values. Section 3.7. - it is necessary to describe the weather conditions on insect observation days, since the attendance of flowers depends very much on the weather - sunny or cloudy, whether it rains, and so on.

Reviewer 2 Report

Comments and Suggestions for Authors

First, see below for comments regarding the English composition-this will need a major review and revision to make this paper publishable. 

In terms of the scientific approach, the methods are classical but need further refinement in the description in the methods, results, and discussion sections. 

Please see the attached file for full comments on both the science and English.

Comments on the Quality of English Language

This paper needs a thorough review and revision of the English. There are innumerable instances of sentence fragments, run-ons, and dangling clauses. Please ensure that each sentence has a subject and action. Check punctuation marks. 

Reviewer 3 Report

Comments and Suggestions for Authors

Review of Zhang et al. (Iris setosa)

This project strikes me as a worthy contribution to the floral biology of Iris.   The article reflects a great deal of interest, time, effort, attractive photography, and technical ability.    Such actual new data on pollination biology is a welcome contribution to organismal botany.    Although I see it in a generally favorable light, the article  needs work.   

The main problem may be rooted in language/translation, which muddies understanding to the point of  misapplication of key terms.  For example, contrary to the wording, Iris setosa is in no sense dioecious (e.g., lines 25, 257, 467).   The term “hybridization index” may be a mistranslation of “Outcrossing Index (OCI).”     “Vegetative reproduction” (line 494) seems misused.  See “self-incompatibility” apparently misused on line 125 and elsewhere).  Also used confusingly in multiple places:  “parthenogenic.”  And more.  I’m not personally able to comb the ms, for every potentially mis-translated term.  The authors should partner with a fluent English speaking botanist familiar with pollination biology during the revision process.   I’m not familiar with the term “honey guides.”    Should this be nectar guides?   These terminology problems are sufficient to impede a clear reading of the authors’s intents.   Beyond that, there is considerable need for general non-botanical copy-editing due to awkward wordings and other issues (e.g., line 19, 47, 51, 63, 72, 103, 452 etc.)

A second major concern is  that Iris and Iridaceae are famous for their unique and beautiful flowers, and thus their floral biology has a substantial prior literature.  This is under-emphasized in the present paper which in many places elaborates on “textbook” level pre-existing knowledge, reinventing the wheel. If nothing else, the present paper could use Goldblatt & Manning’s massive review, which is rich in floral and pollination biology for Iris and similar genera:   “The Iris Family:  Natural History and Classification” (2008).  It is available in full online:  https://www.google.com/books/edition/The_Iris_Family/ZQjvBMInHWQC?hl=en&gbpv=1&dq=iris+floral+biology&pg=PP1&printsec=frontcover

That book has a discussion of self-incompatibility probably relevant to the present research (see around line 311).

Line 413 states that self-pollination cannot occur. But T5 suggests that it can.   Both are possible, but this needs a little discussion, given that if it can, it probably does even if undetected, and even if rare under “natural” conditions.     Given that Iris is rhizomatous with different shoots in different stages, how about by geitonogamy?

There are places, mainly in the Discussion, where the interwoven prior knowledge and present findings are interwoven confusingly.

Minor editorial assists:

Lines 74, 429:  Replace I with we.

Table 1: multiple typos.

439: typo

In short, I’d suggest a major revision where:

-the paper takes prior literature adequately into account

-the ms. is reworded to eliminate “obvious” and well established knowledge, and focuses more on what it adds new to pollination biology of Iris, especially the interesting and well documented artificial pollination experiments.   These strike me as the precious heart of the paper.

-most importantly, the revision  fixes numerous problems with translations, especially of mis-applied botanical terms

Comments on the Quality of English Language

This needs major improvement, especially for botanical terms.

Round 2

Reviewer 1 Report

Comments and Suggestions for Authors

I thank the author for the attentive attitude to the review and the changes made. The article can be accepted for publication

Author Response

Comments 1: I thank the author for the attentive attitude to the review and the changes made. The article can be accepted for publication.

Response 1: Thanks very much for your kind work and consideration on publication of our paper. On behalf of my co-authors, we would like to express our great appreciation to editor and reviewers.

Thank you and best regards.

Reviewer 2 Report

Comments and Suggestions for Authors

Much improved in all aspects. Only final correction is the following

“-”indicates that the stigma is not receptivity, “+”indicates that the stigma is receptivity, “++”indi-286 cates that the stigma is highly receptivity, “+++”indicates that the stigma is extremely receptivity.

change receptivity to receptive in Table 2 above

Author Response

Comments 1: Thank you so much. The authors have thoughtfully revised the manuscript to address the reviewers' concerns.

Much improved in all aspects. Only final correction is the following

“-”indicates that the stigma is not receptivity, “+”indicates that the stigma is receptivity, “++”indi-286 cates that the stigma is highly receptivity, “+++”indicates that the stigma is extremely receptivity.

change receptivity to receptive in Table 2 above

Response 1: Thank you very much for your hard work and consideration, The publication of our paper. We have revised your comments and, on behalf of my co authors, we would like to express our sincere gratitude to the editors and reviewers.

Thank you and my sincerest greetings.

Reviewer 3 Report

Comments and Suggestions for Authors

The authors revised the manuscript addressing reviewer concerns thoughtfully.

Author Response

(The authors gave the same response as above.)
